# Irisin directly stimulates osteoclastogenesis and bone resorption in vitro and in vivo

Eben G Estell[1]*, Phuong T Le[1], Yosta Vegting[1], Hyeonwoo Kim[2], Christiane Wrann[2,3,4,5], Mary L Bouxsein[6], Kenichi Nagano[7], Roland Baron[7], Bruce M Spiegelman[2], Clifford J Rosen[1]*

[1]Maine Medical Center Research Institute, Scarborough, United States; [2]Dana Farber Cancer Institute, Boston, United States; [3]Cardiovascular Research Center, Massachusetts General Hospital, Boston, United States; [4]Department of Medicine, Harvard Medical School, Boston, United States; [5]Department of Cell Biology, Harvard University Medical School, Boston, United States; [6]Beth Israel Deaconess Department of Orthopedic Surgery, Harvard Medical School, Boston, United States; [7]Harvard School of Dental Medicine, Boston, United States

**Abstract** Irisin, a skeletal-muscle secreted myokine, facilitates muscle-bone crosstalk and skeletal remodeling in part by its action on osteoblasts and osteocytes. In this study, we investigated whether irisin directly regulates osteoclasts. In vitro, irisin (2–10 ng/mL) increased osteoclast differentiation in C57BL/6J mouse bone marrow progenitors; however, this increase was blocked by a neutralizing antibody to integrin $\alpha_V\beta_5$. Irisin also increased bone resorption on several substrates in situ. RNAseq revealed differential gene expression induced by irisin including upregulation of markers for osteoclast differentiation and resorption, as well as osteoblast-stimulating 'clastokines'. Forced expression of the irisin precursor *Fndc5* in transgenic C57BL/6J mice resulted in lower bone mass at three ages and greater in vitro osteoclastogenesis from *Fndc5*-transgenic bone marrow progenitors. This study demonstrates that irisin acts directly on osteoclast progenitors to increase differentiation and promote bone resorption, supporting the tenet that irisin not only stimulates bone remodeling but may also be an important counter-regulatory hormone.

*For correspondence:
eestell@mmc.org (EGE);
rosenc@mmc.org (CJR)

## Introduction

Irisin is a peptide produced by proteolytic cleavage of fibronectin type III domain-containing protein 5 (FNDC5), a membrane-bound protein highly expressed in the skeletal muscle. *FNDC5* expression increases in response to acute bouts of exercise by regulation of PGC-1α, leading to a burst of circulating irisin (*Jedrychowski et al., 2015*). Initially, irisin was described as a hormone that induces thermogenesis in adipose tissue (*Boström et al., 2012*), but more recent studies have shown that it has a potent ability to modulate bone turnover. These effects support the tenet that irisin may be a key mediator of muscle-bone crosstalk during exercise. Initial studies demonstrated that irisin enhanced cortical bone formation and prevented unloading-induced bone loss in vivo and stimulated osteoblasts in vitro (*Colaianni et al., 2014*; *Colaianni et al., 2015*; *Colaianni et al., 2017*). Conversely, genetic deletion of *Fndc5* was separately shown to block resorption-driven bone loss and maintain osteocyte function following ovariectomy; irisin treatment in vitro also prevented osteocyte apoptosis and stimulated sclerostin and RANKL release, key promoters of osteoclastogenesis, through the $\alpha_V\beta_5$ integrin receptor (*Kim et al., 2018*). This study addresses the hypothesis that irisin also directly stimulates osteoclast differentiation and function in vitro and in vivo.

## Results and discussion

First, we used continuous treatment with a range of irisin doses (0, 2, 5, 10, and 20 ng/mL) for 7 days in an in vitro osteoclast differentiation assay using primary marrow hematopoietic progenitors. These doses were based on previous studies establishing the physiologic range of circulating irisin during and after exercise (*Jedrychowski et al., 2015*). We found a qualitative enhancement of both osteoclast number and size (*Figure 1a*), and significant increases in osteoclast number across this dose range (2, 5, and 10 ng/mL: p<0.0001, 20 ng/mL: p=0.044; *Figure 1b*, *Source data 1*). Based on these results, we selected 10 ng/mL irisin to compare continuous (7 days) and transient treatment

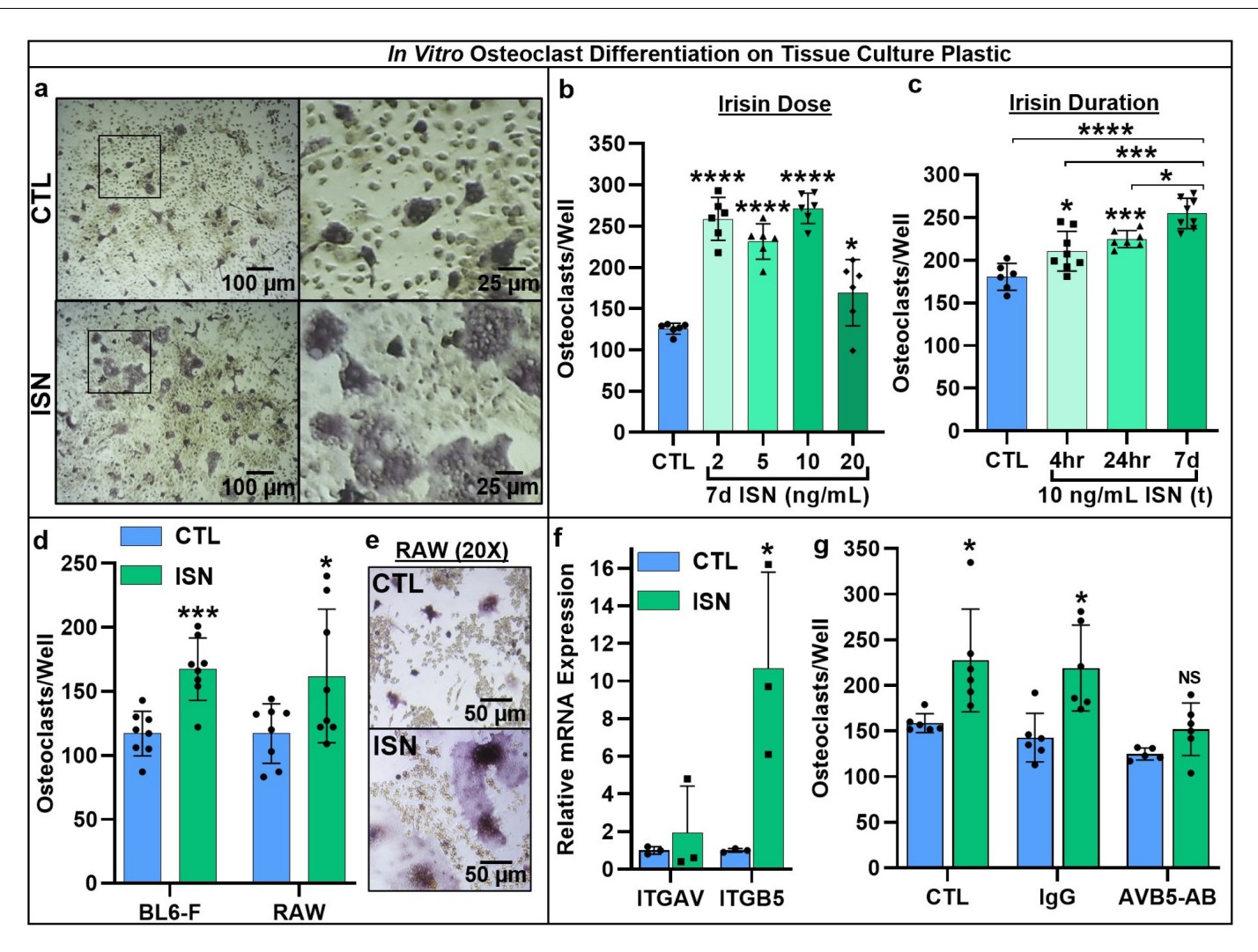

**Figure 1.** Representative 10× and 40× (boxed inset) images of TRAP-positive stained osteoclasts after 7-day differentiation with 10 ng/mL irisin (ISN) or untreated controls (CTL) (**a**). Quantification of osteoclasts per well demonstrating enhanced osteoclastogenesis in response to continuous irisin across a physiologic range of 2–20 ng/mL (**b**, N = 6) and to treatment with 10 ng/mL irisin for only first 4 or 24 hr of culture compared to continuous treatment or CTL (**c**, N = 6–8). Quantification of osteoclasts per well confirming irisin stimulation of osteoclastogenesis with continuous 10 ng/mL treatment across primary murine gender with female BL6 mice, and with the macrophage cell line RAW 264.7 (**d**, N = 8), with representative images of differentiated RAW-derived osteoclasts (**e**). Expression of integrin receptor subunit αV (ITGAV) and β5 (ITGB5) in primary osteoclast cultures normalized to *Hprt* (**f**, N = 3), and quantification of osteoclast per well counts for CTL or continuous 10 ng/mL ISN in the presence of integrin αVβ5 neutralizing antibody (AVB5-AB), an IgG antibody control (IgG), or no antibody (CTL) (**g**, N = 5–6). *p<0.05, **p<0.01, ***p<0.001, ****p<0.0001, not significant (NS) versus CTL within-group or as indicated.

The online version of this article includes the following figure supplement(s) for figure 1:

**Figure supplement 1.** Representative images of in vitro osteoblast cultures with (ISN) or without (CTL) 10 ng/mL irisin during 18-day differentiation.

for the first 4 or 24 hr of culture, and for all further experiments with resorption and gene expression. Both transient treatments led to enhanced osteoclast numbers versus controls (4 hr: p=0.0218, 24 hr: p=0.0008), but continuous irisin resulted in the greatest increase (p<0.0001) and was higher than both 4 hr (p=0.0002) and 24 hr only treatments (p=0.0152; *Figure 1c*, *Source data 1*). The stimulatory effect of continuous 10 ng/mL irisin was then further confirmed in primary hematopoietic cells from both sexes of C57BL/6J mice (p=0.0003) and in the RAW 264.7 macrophage cell line (p=0.0428; *Figure 1d*, *Source data 1*). RAW-derived osteoclasts appeared morphologically similar to primary cells and mirrored observations of qualitatively larger cells with irisin treatment (*Figure 1e*).

As integrins are present on the osteoclast membrane and known to play a role in differentiation (*Duong et al., 2000*; *Yavropoulou and Yovos, 2008*), and earlier work identified integrin $\alpha_V\beta_5$ as a receptor for irisin on osteocytes (*Kim et al., 2018*), we examined the expression of both subunits in osteoclast cultures and found increased relative mRNA expression above controls with irisin treatment ($\alpha_V$: p=0.552, $\beta_5$: p=0.031; *Figure 1f*, *Source data 1*). Blocking integrin $\alpha_V\beta_5$ with a neutralizing antibody (AVB5-AB) resulted in no differences in osteoclast number per well with irisin treatment (10 ng/mL) versus untreated controls (p=0.79) compared to significant increases with an IgG antibody control (p=0.012) or no-antibody conditions (p=0.0295), indicating this integrin acts as the receptor for irisin on osteoclasts (*Figure 1g*, *Source data 1*).

Next, we asked whether irisin-induced osteoclastogenesis led to enhanced bone resorption. Osteoclast differentiation cultures were performed on a variety of native and synthetic substrates with and without continuous 10 ng/mL irisin. TRAP-positive osteoclasts were observed in situ on dentin slices after 7 days (*Figure 2a*, left), and subsequent toluidine blue staining revealed a qualitative increase in resorption pit area on the surface with irisin treatment (*Figure 2a*, right). Irisin significantly increased osteoclast numbers on dentin (p=0.013), as well as total resorption area (p=0.045). However, when normalized by osteoclast number, resorption was not significantly different (p>0.99), indicating a dominant effect of cell number in increasing total resorption (*Figure 2b*, *Source data 1*). Irisin enhancement of total resorption was further confirmed via the Corning OsteoAssay, with a similar significant increase in total resorption area (p=0.048; *Figure 2c*, *Source data 1*). Release of carboxy-terminal collagen crosslinks (CTX) from osteoclast cultures on collagen substrates was measured to assess the effect of transient irisin treatment on resorption at early stages in culture and was significantly increased with both continuous irisin treatment for 3 days (p=0.0153) and for the first 24 hr only (p<0.0001), with this transient treatment also resulting in significantly higher resorption than continuous treatment (p=0.0041; *Figure 2d*, *Source data 1*).

To investigate the key signaling pathways influenced by the action of irisin on the osteoclast, we performed an unbiased analysis of RNA sequencing (RNAseq) data from in vitro osteoclast cultures after 7 days of continuous 10 ng/mL irisin treatment, which demonstrated a qualitatively differential gene expression pattern compared to untreated controls, as typified by hierarchical clustering, volcano plot, and principal component analysis (*Figure 3a*). Irisin treatment significantly increased the expression of resorption markers *Adamts5* (p=0.0381) and *Loxl2* (p=0.009), and markers for secreted clastokines known to stimulate osteoblasts: *Postn* (p=0.0002), *Igfbp5* (p=0.03), *Tgfb2* (p=0.0073), and *Sparc* (p=0.0365). Significant decreases in the expression of macrophage markers *Mst1r* (p=0.0416) and *Itgax* (p<0.0001) and the lymphocyte markers *Cd72* (p=0.0086), *Slamf8* (p=0.0052), and *H2-aa* (p=0.0017) indicated a preferential shift of the hematopoietic progenitor lineage toward osteoclast differentiation (*Source data 1*). RT-qPCR further showed that irisin significantly increased the expression of key differentiation markers, namely, *Rank*, the receptor for RANKL (p=0.019), *c-src* (p=0.008), *Fam102a* (p=0.039), *Nrf2* (p=0.0008), and the osteoclast fusion markers *Dcstamp* (p=0.0039) and *Atp6vod2* (p=0.012). Other early markers of osteoclast differentiation such as *Cfos* (p=0.41), *Itgb3* (p=0.09), *Nfatc* (p=0.11), *Rela* (p=0.12), and *Rgs12* (p=0.09) were slightly but not significantly increased. Significant increases were confirmed for both key resorption markers: *Acp5* (p=0.012) and *Loxl2* (p=0.035), and clastokine markers: *Sparc* (p=0.009) and *Wnt10a* (p=0.047; *Figure 3b*, *Source data 1*).

Next, we turned to a genetic model of forced expression of *Fndc5* in C57BL/6J mice using the *McK* muscle-specific promoter. RT-qPCR analysis of the whole tibia demonstrated a significant increase in *Fndc5* expression at both 4.5 months (p=0.000082) and 13 months (p=0.0041), indicating promoter activity in the bone marrow in addition to the muscle (*Figure 4a*, *Source data 1*). Markedly lower cortical bone area was observed by μCT at younger ages in *Fndc5*-transgenic (TG) mice

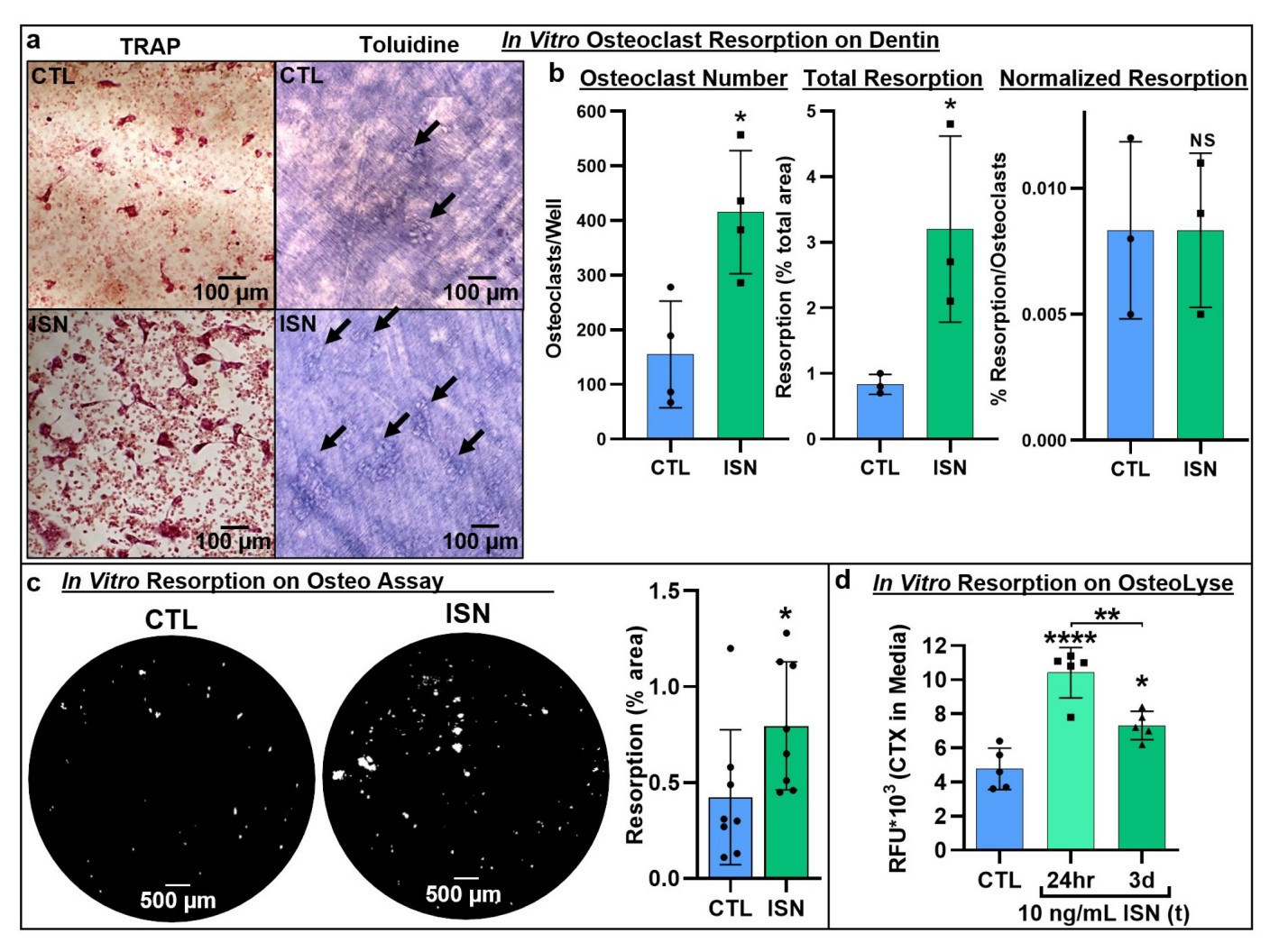

**Figure 2.** Representative images of TRAP-positive osteoclasts on dentin after 7-day osteoclast culture (left), and underlying resorption pits (arrows) stained with toluidine blue dye after decellularization via sonication, demonstrating increased resorption with irisin treatment (ISN) versus untreated controls (CTL) (a). Quantification of osteoclast number, total resorption area, and resorption normalized to osteoclast number for dentin cultures (b, N = 3–4). Confirmation of irisin stimulation of resorption with Corning OsteoAssay resorbable calcium phosphate substrate, representative full-diameter images of 96-well plates with binary threshold to visualized resorption pits on von Kossa-stained substrate after 7-day osteoclast culture for ISN versus CTL, with quantification of total resorption by percent area (c, N = 8). Irisin stimulation of early-stage resorption with transient treatment for the first 24 hr alone, versus continuous ISN and CTL with quantification of resorption determined by ELISA of CTX release into media collected at day 3 of culture (d, N = 5). *p<0.05 versus CTL, **p<0.01, ****p<0.0001, not significant (NS) versus CTL within-group or as indicated.

compared to wild type (WT) (2 and 4.5 months), with significant decreases in trabecular bone volume fraction at 2 (p<0.0001) and 4.5 months (p=0.0003) and in cortical thickness at 2 months (p<0.0001; *Figure 4b*, *Source data 1*). Dynamic histomorphometry revealed smaller bones and similar decreases in overall bone volume fraction and trabecular thickness that persisted from 2 to 13 months (*Figure 4c*, *Source data 1*). Significant decreases in both the number of osteoblasts per bone perimeter (p=0.05) and bone formation rate (p=0.0046) were observed only at 2 months, and not at 13 months. However, in vitro osteoblast differentiation from bone marrow mesenchymal cells harvested at 2–3 months did not differ by genotype (*Figure 4—figure supplement 1*). Osteoclast numbers per bone perimeter were slightly increased in Fndc5-transgenic mice versus wild type across 2–13 months but did not reach statistical significance (*Figure 4c*). However, primary bone marrow progenitors isolated from *Fndc5*-transgenic mice at 2–3 months demonstrated markedly

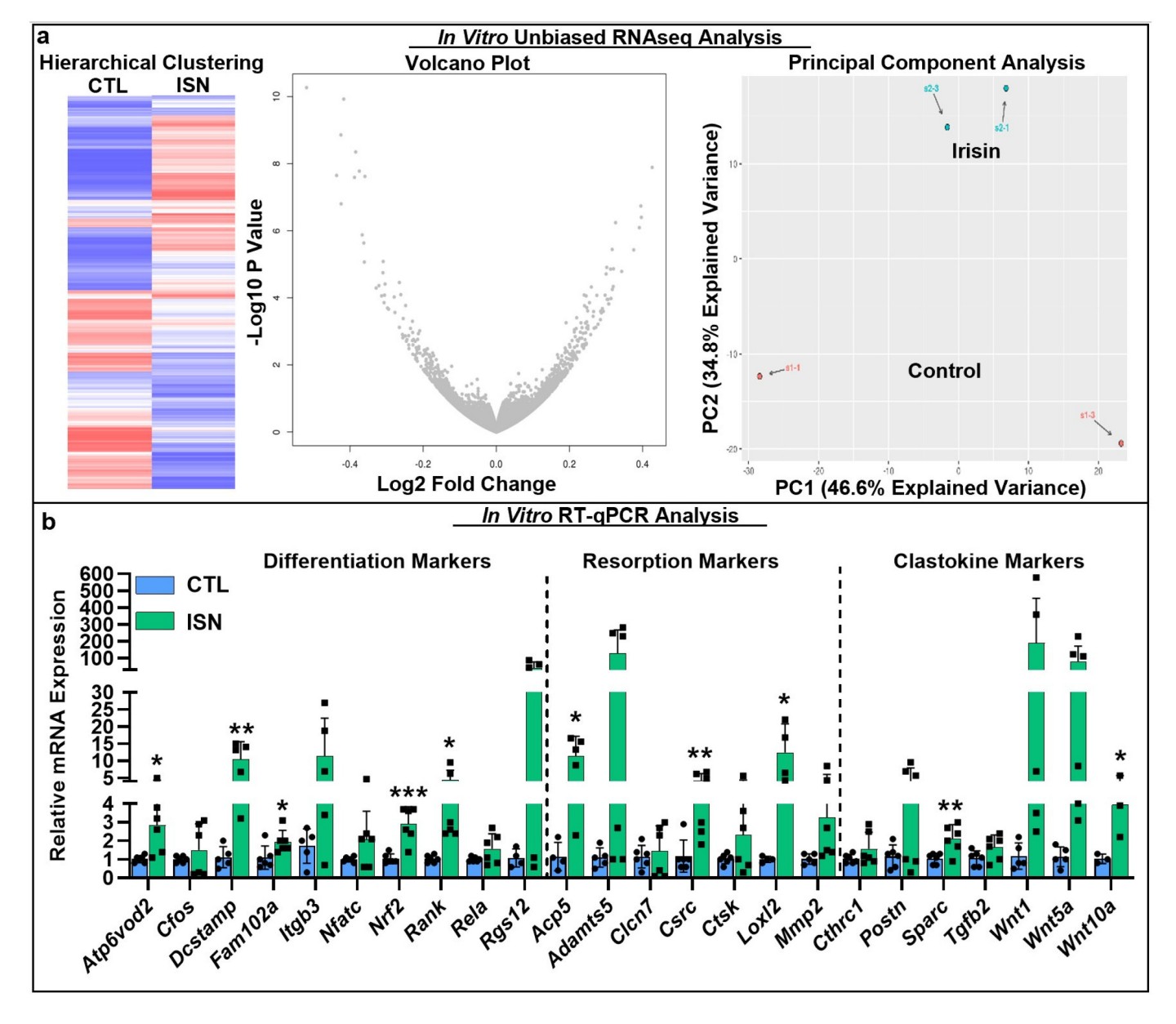

**Figure 3.** RNAseq analysis of differential gene expression pattern induced by continuous 10 ng/mL irisin treatment (ISN) compared to untreated controls (CTL) at day 7, as typified by representative sample hierarchical clustering, volcano plot, and principal component analysis (a, N = 2). Relative mRNA expression quantified by RT-qPCR of markers for osteoclast differentiation, resorption, and clastokines in irisin-treated osteoclasts (ISN) compared to untreated controls (CTL), normalized to *Hprt* expression (b, N = 3–6). *p<0.05, **p<0.01, ***p<0.001 versus CTL within gene.

greater osteoclastogenic potential compared to the wild type, yielding significantly higher numbers of osteoclasts that were qualitatively larger than controls during in vitro differentiation (p<0.0001; *Figure 4d*, *Source data 1*).

This study demonstrates that irisin plays an important role in regulating bone remodeling not only by stimulating osteoblasts and osteocytes as previously described, but by also directly acting on osteoclasts to promote differentiation and resorption. This stimulatory effect was observed across multiple experiments with primary murine progenitors and the RAW 264.7 macrophage cell line and occurred with either intermittent or continuous irisin exposure across a range of physiologic concentrations previously reported in humans (*Jedrychowski et al., 2015*). Analogous to its action on

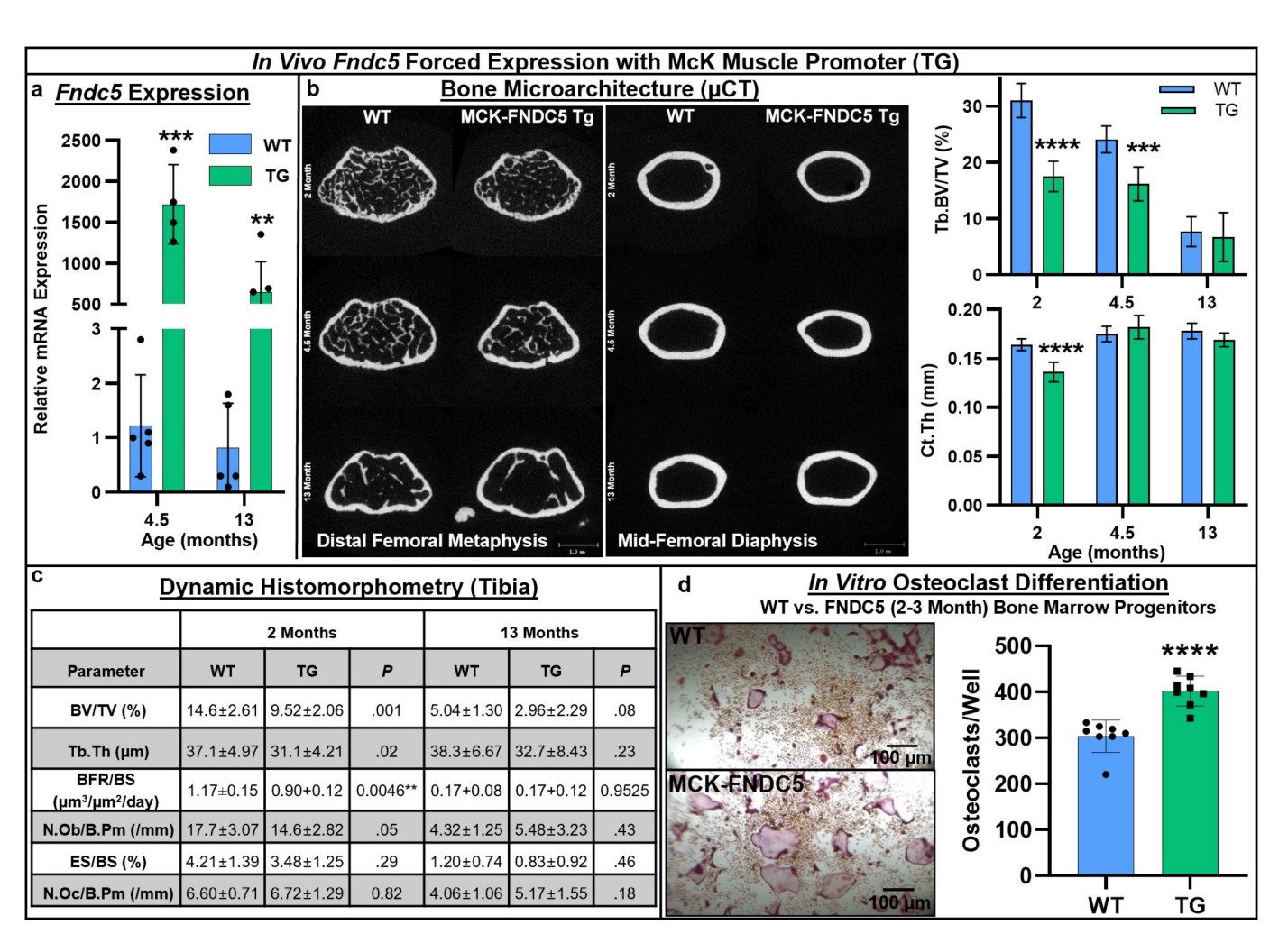

**Figure 4.** Skeletal phenotype and osteoclastogenic potential of *Fndc5* forced expression with McK muscle promoter mice (MCK-FNDC5, TG) compared to wild type C57BL/6J controls (WT). Whole-bone gene expression from tibia shows increased *Fndc5* expression out to 13 months (**a**, N = 3–5). Representative transverse slice images of trabecular bone at the midpoint of the distal femoral metaphysis (left) and cortical bone at the mid-diaphysis (right) demonstrates reduced cortical bone area in TG versus WT, especially at early ages, with trabecular BV/TV significantly lower at 2 and 4.5 months, and cortical thickness significantly lower at 2 months (**b**, N = 6). Tibial histomorphometry at 2 and 13 months demonstrated reduction of bone volume fraction at both time points, with slight increase in osteoclast numbers in tibia at 13 months (**c**, N = 6). In vitro MCSF/RANKL-induced osteoclast differentiation from bone marrow progenitors at 2–3 months of age yielded higher osteoclast numbers in TG versus WT (**d**, N = 8). ***p<0.001, ****p<0.0001 versus WT.

The online version of this article includes the following figure supplement(s) for figure 4:

**Figure supplement 1.** Representative images of in vitro osteoblast cultures from wild type (WT) or *Fndc5*-transgenic bone marrow progenitors after 18-day differentiation.

osteocytes, we confirmed the expression of integrin subunits $\alpha_V$ and $\beta_5$ on osteoclasts and identified it as a likely receptor for irisin, particularly since blocking this receptor complex with a neutralizing antibody completely suppressed the stimulatory effect of irisin on osteoclastogenesis (*Figure 1*). Furthermore, we found that both recruitment and differentiation of more osteoclast progenitors with irisin treatment appeared to be driving factors of enhanced bone resorption, based on in situ studies on native dentin as well as synthetic calcium phosphate and collagen substrates (*Figure 2*). Using unbiased RNAseq analysis and qRT-PCR of irisin-treated osteoclasts, we noted that some markers of early osteoclast differentiation, nuclear fusion markers, and enzymes related to bone resorption

were upregulated, matching the functional effects observed in vitro. In addition, several osteoclast-secretory factors known to stimulate osteoblasts were significantly upregulated, suggesting that the direct actions of irisin on this cell type may have further impact on cell signaling to enhance coupled remodeling (*Figure 3*).

To confirm the capacity of irisin to impact osteoclast function in vivo, we employed a genetic strategy with chronically forced expression of *Fndc5* using the muscle-specific *Mck* promoter. In that mouse model, we demonstrated that high levels of irisin were expressed in the whole bone marrow at both ages, and this was associated with low bone mass. The relative contribution of osteoclasto-genesis to the skeletal phenotype cannot be resolved in our genetic model. While early decreases in osteoblast numbers and bone formation rate were observed by histomorphometry at 2 months of age in the transgenic mouse, these metrics became equivalent to controls at 13 months, whereas a trend for increased osteoclast numbers was noted at 13 months. The in vitro differentiation cultures from the *Fndc5*-transgenic marrow progenitors from 2- to 3-month-old mice provide further support-ive evidence for the action of irisin on the osteoclast. Bone marrow mesenchymal progenitors from *Fndc5*-transgenic mice at 2–3 months of age were not different from the wild type in regard to the pace or quantitation of osteoblast differentiation or mineralization. On the other hand, in vitro osteo-clastogenesis from transgenic hematopoietic progenitors was markedly enhanced compared to con-trol mice (*Figure 4*). Additionally, we have found that continuous exogenous treatment with 10 ng/mL irisin used to stimulate in vitro osteoclast differentiation from wild type progenitors had no observable effect on osteoblast differentiation or mineralization (*Figure 1—figure supplement 1*). Taken together, our results demonstrate a clear action of irisin on the osteoclast, but further suggest that in this forced expression model, irisin may act at early stages to reduce bone mass by suppress-ing osteoblast-mediated bone formation, while the later and persistent effects on bone turnover may be more focused on the osteoclast.

While our observations of irisin's effects on osteoblastogensis differ from other published studies (*Colaianni et al., 2014*), it is important to note that both dose and duration of irisin exposure may be critical determinants of the skeletal response and hence be responsible for phenotypic differen-ces. For example, from our gene expression studies, it is clear that the activity of the *Mck* promoter for *Fndc5* is very high at 2 months but waned with age, albeit still maintaining relatively increased levels (*Figure 4a*). This exposure difference may impact skeletal cell responsiveness as noted by an earlier study demonstrating suppressed osteoclastogenesis in vitro with supraphysiologic levels of irisin (*Marino et al., 2014*). Similarly, we found that 20 ng/mL irisin had little effect on osteoclast number in vitro (*Figure 1b*), and initial experiments with doses of irisin equal or exceeding 100 ng/mL actually showed reduced osteoclast numbers (data not shown). Our previous work employed mass spectrometry to demonstrate that irisin concentrations in humans are approximately 2–4 ng/mL (*Jedrychowski et al., 2015*). As direct quantification of irisin protein levels via ELISA remains a challenge for the field, we used this range measured via mass spectrometry to guide our selection of irisin doses. While we primarily employed a concentration of 10 ng/mL to represent a robust but physiologically relevant in vitro stimulus, we also demonstrated that even low concentrations down to 2 ng/mL significantly increase osteoclast number. Thus, both dosing and timing relative to devel-opmental age may be determinants of irisin's function as a dynamic signaling molecule in the skele-ton, and its actions on specific bone cell types. It should be noted that the relatively narrow dose range for stimulating osteoclastogenesis could have physiologic implications since exercise induces rapid changes in serum calcium (*Ng et al., 2018*). If irisin targets both osteoclasts and osteocytes, it may serve as another counter-regulatory hormone to maintain serum calcium during the early phases of exercise much like parathyroid hormone (*Ng et al., 2018*).

Taken together, our studies provide more evidence that irisin mediates muscle-bone crosstalk by regulating bone remodeling. While further studies are necessary to fully elucidate irisin's mechanism of action on the osteoclast and to resolve contrasting cellular effects observed between cell types and experimental models, it remains clear that this myokine is a potent regulator of bone remodel-ing that is able to both act directly on all key cell types of the bone remodeling unit, and potentially modulate signaling between them.

# Materials and methods

**Key resources table**

| Reagent type (species) or resource | Designation | Source or reference | Identifiers | Additional information |
|---|---|---|---|---|
| Strain, strain background (*Mus musculus* male/female) | C57BL/6J | Jackson Laboratories | IMSR Cat# JAX:000664, RRID:IMSR_JAX:000664 | Wild type mouse line |
| Cell line (*Mus musculus*) | RAW 264.7 | ATCC | TIB-71 | Cryopreserved monocyte cell line |
| Antibody | Anti-Integrin aVb5 (Mouse monoclonal) | Abcam | ab78289, RRID:AB_1566022 | Neutralizing antibody (1:1000) |
| Peptide, recombinant protein | Irisin | Lake Pharma *Kim et al., 2018* | N/A (custom) | 10 his-tag recombinant from HEK 293 cells |
| Peptide, recombinant protein | RANKL, recombinant human | PeproTech | 310–01 | Osteoclast growth factor |
| Peptide, recombinant protein | M-CSF, recombinant murine | PeproTech | 315–02 | Osteoclast growth factor |
| Commercial assay or kit | Acid Phosphatase Kit | Sigma-Aldrich | 387A | TRAP assay |
| Commercial assay or kit | Corning Osteo Assay | VWR | 89184–614 | Resorption assay |
| Commercial assay or kit | OsteoLyse Assay Kit | Lonza | PA-1500 | Resorption assay |
| Software, algorithm | Prism 8 | Graphpad | RRID:SCR_002798 | Statistics/graphing software |
| Software, algorithm | FIJI | NIH | RRID:SCR_002285 | Image analysis software |

## Primary osteoclast culture

Primary murine osteoclasts were differentiated and cultured in vitro by the following methods. Bone marrow was collected via centrifugation from the femur and tibia of 8-week-old male C57BL6/J mice and cultured in αMEM (VWR, Radnor, PA) supplemented with 10% fetal bovine serum VWR) and 1% Pen-Strep (VWR). After 48 hr, non-adherent hematopoietic progenitor cells were removed and plated at $1.563 \times 10^5$ cell/cm$^2$ in 96-well tissue culture plates for cell counting or in 12-well tissue culture plates for RNA extraction. Osteoclast differentiation was stimulated by supplementation with 30 ng/mL M-CSF (PeproTech, Rocky Hill, NJ) and 100 ng/mL RANKL (PeproTech; *Marino et al., 2014*), with 200 µL or 2 mL media refreshed at days 3 and 5 after plating for 96- and 12-well plates, respectively. Irisin was produced in HEK 293 cells as a 10 His-tag recombinant, via previously established protocols (Lake Pharma, Hopkinton, MA; *Kim et al., 2018*), and supplemented continuously in the media at 10 ng/mL, or as otherwise indicated.

## Primary source gender and cell line confirmations

We used 8-week-old male C57BL6/J mice as the primary cell source for all experiments unless otherwise stated. Confirmation of irisin effect on osteoclastogenesis was also established in female C57BL6/J mice to compare gender among this wild type murine primary source of progenitors, which were cultured and counted as described. Additionally, the RAW 264.7 macrophage cell line (ATCC TIB-71, Manassas, VA) was employed as a non-primary cell source, following previously published protocols for osteoclast differentiation from this cell line (*Ng et al., 2018*; *Bharti et al., 2004*). Briefly, RAW cells were played at a lower density in 96-well plates of $6 \times 10^3$ cells/well and cultured as described, but for the exclusion of MCSF in the media. The RAW 264.7 cell line is not included in the database of commonly misidentified cell lines maintained by the International Cell

Line Authentication Committee; identity authentication via STR profiling and a negative mycoplasma contamination test were performed by the supplier ATCC.

## Osteoclast counts

At day 7, 96-well plates were fixed in 10% formalin and stained for TRAP (Acid Phosphatase Kit, Sigma-Aldrich, St Louis, MO) to visualize and count mature osteoclast numbers, where a TRAP-positive cell with three or more nuclei was defined as an osteoclast. Initial counting was performed via manual counting on an inverted microscope, with confirmative counts performed by manually counting blinded copies of composite images of each sample in ImageJ (Blind Analysis, Labcode).

## Integrin antibody blocking

A separate experiment employed the culture and counting methods described above for irisin treatment in combination with a neutralizing antibody for integrin $\alpha_V\beta_5$ (Anti-Integrin aVb5 antibody, Abcam, Cambridge, UK) and an IgG control (Mouse IgG1 Isotype Control, R and D Systems, Minneapolis, MN), each supplied continuously in the media during the 7-day culture at 0.9 µg/mL.

## Resorption assays

Multiple resorption assays were utilized to characterize and confirm the effect of irisin treatment on osteoclast resorptive capacity. The primary approach employed decellularized dentin slices as a native bone substrate. Hematopoietic progenitors were plated on the bone slices at $3 \times 10^5$ cell/cm$^2$ in a 50 µL place on the top of the slice in a 10 mm Petri dish and incubated for 30 min to facilitate cell adhesion to the substrate alone. Slices were then moved into 96-well plates and cultured as described above. At day 7, dentin slices were TRAP stained and imaged as described for osteoclast counts. TRAP+ osteoclasts were visualized on the dentin surface via a top-down microscope used for histomorphometry analysis, with bright light illumination provided from below, through the thin dentin slice. To image the underlying resorption pits, the dentin slices were then sonicated briefly to removed cells and stained with toluidine blue to visualize resorption pits by previously published methods (*Vesprey and Yang, 2016*) on the same microscope. Briefly, each slice was placed face-down on a 20 µL drop of 1% toluidine blue solution (in 1% sodium borate 10-hydrate solution with distilled water) for 4 min and then rinsed and allowed to air dry before imaging and manual calculation of total pit areas for blinded images in ImageJ. Toluidine-stained pits were clearly identifiable on the dentin surface as rounded features with dark edges and were manually traced in Image J to generate binary thresholded masks from which the total area was calculated. Confirmation experiments were performed with the Osteo Assay resorption assay (Corning Inc, Corning, NY), whereby osteoclasts were cultured on the substrate treated 96-well plates by the previously described methods, then removed with 10% bleach and the substrate was stained with von Kossa stain (Sigma-Aldrich). Imaging of the wells on a dissecting scope with backlighting allowed visualization of pits, and automated area calculations based on binary thresholded masks of the images. To determine earlier time-point resorption, the OsteoLyse assay (Lonza, Basel, Switzerland) was employed, utilizing the same osteoclast culture methods on a collagen substrate, whereby detection of carboxy-terminal collagen crosslinks (CTX) allows for relative quantification of degraded collagen as an indicator of resorptive activity. Aliquots of the media at day 3 in culture were analyzed via ELISA for relative fluorescence indicative of CTX release and normalized to undifferentiated and no-cell controls.

## Gene expression analysis

Total RNA was isolated from osteoclast cells cultured with or without 10 ng/mL irisin for 7 days with the Trizol reagent (ThermoFisher, Waltham, MA) and RNeasy mini kit (Qiagen, Hilden, Germany), with mRNA enrichment from 100 ng of total purified RNA and Illumina sequencing libraries preparation performed using Kapa stranded mRNA Hyper Prep (Roche Sequencing Solutions, Pleasanton, CA). Gene libraries were multiplexed in an equimolar pool and were sequenced on an Illumina NextSeq 500 with single-end 75 bp reads. Raw reads were aligned to the UCSC mm10 reference genome using a STAR aligner (*Dobin et al., 2013*; version STAR_2.4.2a), and raw gene counts were quantified using the quantMode GeneCounts flag. Differential expression testing was performed using

Limma (*Ritchie et al., 2015*) and DESeq2 (*Love et al., 2014*). RNAseq analysis was performed using the VIPER snakemake pipeline (*Cornwell et al., 2018*). Follow-up RT-qPCR was performed on RNA from a separate set of osteoclasts for markers identified via RNAseq and additional targets for differentiation, resorption, and clastokines with primer sequences obtained PrimerBLAST (NCBI-NIH), using reverse transcriptase kit (Qiagen) and AzuraQuant Green Fast PCR Mix (Azura Genomics, Rynam, MA) with an IQ PCR detection system (Bio-Rad, Hercules, CA). Gene expression data were analyzed via the comparative Ct method, utilizing *Hprt* as the housekeeping gene and normalizing by untreated control mean Ct.

## Forced expression of Fndc5 in the murine muscle

Transgenic mice with muscle-specific forced expression of Fndc5 were generated on a C57BL/6J background utilizing a muscle creatine kinase (MCK) promoter as previously described for PGC1-a overexpression[17], targeting the coding sequence of the mouse Fndc5 gene. Transgenic mice were backcrossed to the C57BL/6J background prior to experiments, and non-transgenic littermate controls were utilized for skeletal phenotyping (*Lin et al., 2002*). Microarchitecture of distal trabecular bone and midshaft cortical bone was analyzed at 2, 4.5, and 14 months by µCT and at 2 and 13 months by dynamic histomorphometry, with measures performed and analyzed according to standard nomenclatures.

The µCT was performed with a high-resolution desktop micro-tomographic imaging system (µCT40, Scanco Medical AG, Brüttisellen, Switzerland) to assess trabecular bone microarchitecture and cortical bone morphology in the femoral distal metaphysis and mid-diaphysis, respectively. Scans were acquired with a 10 µm$^3$ isotropic voxel size, 70 kVP, 114 mAs, 200 ms integration time, subjected to Gaussian filtration and segmentation. Transverse distal femur slices were evaluated in a region beginning 200 µm above the peak of the distal growth plate and extending proximally 1500 µm (150 slices). The trabecular bone region was identified by semi-manually tracing the region of interest with assistance from an auto-contouring software algorithm. Images were thresholded with an adaptive-iterative algorithm (*Ridler, 1978*; *Meinel et al., 2005*; *Rajagopalan et al., 2005*). The average adaptive-iterative threshold of the 4.5-month *Fndc5*-transgenic mice (232 mgHA/cm$^3$) was used for all groups as the threshold to segment bone from soft tissue. Morphometric variables were computed from binarized images using established direct 3D techniques independent of assumptions about the underlying structure (*Hildebrand et al., 1999*; *Hildebrand and Rüegsegger, 1997*; *Arlot et al., 2008*). Cortical bone parameters were assessed from 50 transverse µCT slices obtained at the femoral mid-diaphysis using a 10 µm isotropic voxel size. The ROI, including all pixels within the outermost edge of the cortex, was subjected to Gaussian filtration and segmented using a fixed threshold of 700 mgHA/cm$^3$.

Dynamic histomorphometry was performed on fixed right tibiae, dehydrated with acetone, and embedded in methylmethacrylate. Undecalcified 4-µm-thick sections were obtained by microtome and stained with von Kossa, with consecutive sections left unstained fluorescence label analysis and stained with toluidine blue for identification of osteoblasts and osteoid. Additional consecutive sections were stained with tartrate-resistant acid phosphatase (TRAP) and counterstained with toluidine blue for identification of osteoclasts. Histomorphometric analysis was performed in the proximal tibia under 200 magnification in a 0.9 mm by 1.3 mm wide region 200 µm away from the growth plate using OsteoMeasure (Osteometrics Inc, Decatur, GA). Structural parameters were obtained from three consecutive sections and averaged. Structural, dynamic, and cellular parameters were calculated and expressed according to the standardized nomenclature (*Dempster et al., 2013*).

Additional femur and tibia were pulverized at 4.5 and 13 months with RNA extraction and subsequent gene expression analysis was performed via previously described protocols. Bone marrow isolation and in vitro osteoclast cultures were performed via previously described protocols. In vitro osteoblast cultures were performed by previously established protocols from the same adherent mesenchymal cell population of the same bone marrow isolations from wild type and transgenic mice used for osteoclast cultures. Briefly, after removal of non-adherent hematopoietic progenitors from culture flask for osteoclast culture, adherent mesenchymal progenitors were trypsinized and plated at 1 10$^6$ cell/well in 6-well plates with three replicates per group, and cultured in αMEM supplemented with 10% fetal bovine serum and 1% Pen-Strep until confluent. Confluent cells were then cultured for 18 days, supplemented with β-glycerol phosphate

(Sigma-Aldrich) and ascorbic acid (Sigma-Aldrich), and stained for cell proliferation with crystal violet stain (Sigma-Aldrich), differentiation with an alkaline phosphatase kit (Sigma-Aldrich), and mineralization with von Kossa staining (silver nitrate, Sigma-Aldrich). All experiments were conducted with six age-matched female mice.

## Experimental design and data analysis

Isolation of primary murine bone marrow was conducted by pooling tissue from the maximum available number of same-gender littermates (N = 3–5). For in vitro cultures, the adequate number of biological replicates (replicate wells in a tissue culture plastic plate, or slices of dentin) was determined via power analyses based on preliminary data ($\alpha$ = 0.05, power = 0.8) as six, and so osteoclast counting and resorption experiments were conducted in triplicate with representative experiments shown, with 6–8 replicate wells per group in each experiment (*Figure 1b,c,d,e,g*, *Figure 2c,d*, *Figure 4d*). Similarly, gene expression analysis via RT-qPCR was conducted for duplicate repeat experiments with three biological replicates per group and two technical replicates (replicate wells read per sample and averaged), with pooled representative data for a sample number of six (*Figure 1f*, *Figure 3b*, *Figure 4a*). Due to limitations in the availability of dentin slices, this resorption experiment was conducted once with a sample size of five per group (*Figure 2a*), and was thus followed up with additional commercially available resorption assays (*Figure 2c,d*). Similarly, RNA sequencing analysis was performed on a separate experimental set with three biological replicates per group (*Figure 3a*). For characterization of the *Fndc5*-transgenic mouse, power analyses based on previous outcome metrics from histomorphometry and gene expression in the *Fndc5*-null mouse experiments (*Kim et al., 2018*; $\alpha$ = 0.05, power = 0.8) indicated an adequate sample size of six mice per group, which was employed for in vivo characterization of bone properties based on the availability of same-gender age-matched mice (*Figure 4b,c*), while in vitro culture of osteoclast progenitors from these mice were carried out with bone marrow isolates from maximal number of same-gender littermates (N = 3–5) and eight replicate wells of osteoclast differentiation cultures per group (*Figure 4d*).

For statistical analysis, outlier identification was first performed via Grubb's test with $\alpha$ = 0.05. Based on the recommended guidelines for analysis, comparisons between two groups alone (irisin versus control osteoclast counts, resorption, gene expression) was conducted via unpaired, two-tailed t-test, $p < 0.05$, while multiple group comparisons were made via ordinary one-way (irisin dose and duration versus control) or two-way (antibody/irisin treatment osteoclast counts) ANOVA with Tukey post-hoc analysis and $p < 0.05$. RNA sequencing data analysis was performed as described above with statistical significance via the Wald test with the Benjamini-Hochberg adjustment. Quantitative data are represented graphically as mean ± standard deviation with individual values overlaid.

## Acknowledgements

This work was funded in part by NIH/NIAMS F32AR077382, NIH U19AG060917, NIH U54GM115516-01A1, NIH/NIDDK R01 DK112374, and NIH/NIGMS 1P20GM121301. We thank Zach Herbert, Maura Berkeley, and Andrew Caruso from the Molecular Biology Core Facilities at the Dana-Farber Cancer Institute for RNAseq. We thank Dr Eric Olson of UT Southwestern for providing the transgenic *Fndc5* mice.

## Additional information

### Competing interests

Clifford J Rosen: Senior editor, *eLife*. The other authors declare that no competing interests exist.

### Funding

| Funder | Grant reference number | Author |
| --- | --- | --- |
| National Institute of Arthritis and Musculoskeletal and Skin | F32AR077382 | Eben G Estell Roland Baron |

| | | Bruce M Spiegelman Clifford J Rosen |
| --- | --- | --- |
| Diseases | | |
| National Institute on Aging | U19AG060917 | Eben G Estell Clifford J Rosen |
| National Institute of General Medical Sciences | U54GM115516-01A1 | Clifford J Rosen |
| National Institute of Diabetes and Digestive and Kidney Diseases | R01DK112374 | Clifford J Rosen |
| National Institute of General Medical Sciences | 1P20GM121301 | Clifford J Rosen |

The funders had no role in study design, data collection and interpretation, or the decision to submit the work for publication.

## Author contributions
Eben G Estell, Conceptualization, Data curation, Formal analysis, Funding acquisition, Validation, Investigation, Visualization, Methodology, Writing - original draft, Project administration, Writing - review and editing; Phuong T Le, Conceptualization, Supervision, Validation, Methodology; Yosta Vegting, Conceptualization, Validation, Methodology; Hyeonwoo Kim, Conceptualization, Resources, Investigation, Writing - review and editing; Christiane Wrann, Kenichi Nagano, Data curation, Formal analysis; Mary L Bouxsein, Conceptualization, Data curation, Formal analysis, Writing - review and editing; Roland Baron, Conceptualization, Resources, Supervision, Writing - review and editing; Bruce M Spiegelman, Conceptualization, Resources, Writing - review and editing; Clifford J Rosen, Conceptualization, Resources, Formal analysis, Supervision, Funding acquisition, Project administration, Writing - review and editing

## Author ORCIDs
Eben G Estell https://orcid.org/0000-0001-8319-6886

## Ethics
Animal experimentation: All wild type primary cell progenitors for osteoclast cultures with exogenous irisin were obtained from tissue harvested from male and female C57BL6/J mice housed and treated under a protocol approved by the Maine Medical Center Research Institute IACUC (1914).

## Decision letter and Author response
Decision letter https://doi.org/10.7554/eLife.58172.sa1
Author response https://doi.org/10.7554/eLife.58172.sa2

# Additional files

## Supplementary files
• Source data 1. Source data tables for figures.

• Supplementary file 1. Primer Sequence List. Primer sequences used for quantitative RT-PCR on RNA isolated from irisin treated and control primary osteoclast cultures.

• Transparent reporting form

## Data availability
All data generated or analysed during this study are included in the manuscript figures and Source data 1, with the RNAseq dataset uploaded to the Dryad Digital Repository under 'Eben Estell, Phuong T Le, Yosta Vegting, Hyeonwoo Kim, Christiane Wrann, Mary L Bouxsein, Kenichi Nagano, Roland Baron, Bruce M Spiegelman, Clifford Rosen (2020). Irisin directly stimulates osteoclastogenesis and bone resorption in vitro and in vivo: RNAseq dataset [Dataset]. Dryad Digital Repository, doi: 10.5061/dryad.hqbzkh1dr'.

The following dataset was generated:

| Author(s) | Year | Dataset title | Dataset URL | Database and Identifier |
|---|---|---|---|---|
| Estell E, Le PT, Vegting Y, Kim H, Wrann C, Bouxsein ML, Nagano K, Baron R, Spiegelman BM, Rosen C | 2020 | Irisin directly stimulates osteoclastogenesis and bone resorption in vitro and in vivo: RNAseq dataset | https://doi.org/10.5061/dryad.hqbzkh1dr | Dryad Digital Repository, 10.5061/dryad.hqbzkh1dr |

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
