## [Decision Letter]

**Acceptance summary:**

The reviewers determined that is a well-designed study of the effects of Irisin on osteoclastogenesis. Using a combination of in vitro, RNAseq, and in vivo studies (transgenic mouse model over-expressing Fndc5, an upstream regulator of Irisin in response to exercise), this work provides strong evidence that, in addition to its known effects on osteoblasts and osteocytes, irisin directly regulates osteoclasts and that the hematopoietic progenitors were biased towards osteoclast differentiation.

**Decision letter after peer review:**

Thank you for submitting your article "Irisin directly stimulates osteoclastogenesis and bone resorption in vitro and in vivo" for consideration by *eLife*. Your article has been reviewed by three peer reviewers, and the evaluation has been overseen by a Reviewing Editor and Kathryn Cheah as the Senior Editor. The following individuals involved in review of your submission have agreed to reveal their identity: Warren Grayson (Reviewer #2); Sundeep Khosla (Reviewer #3).

The reviewers have discussed the reviews with one another and the Reviewing Editor has drafted this decision to help you prepare a revised submission.

Summary:

This is a well-designed study of the effects of Irisin on osteoclastogenesis that reports the effects of dosing and timing of Irisin on osteoclast number, morphology, and integrin α_V_β_5_ expression in vitro. Using a combination of in vitro, RNAseq, and in vivo studies (transgenic mouse model over-expressing Fndc5, the precursor protein of irisin), the authors provide strong evidence that, in addition to its known effects on osteoblasts and osteocytes, irisin directly regulates osteoclasts. The authors also found that the hematopoietic progenitors were biased towards osteoclast differentiation along with an increase in some resorption markers and clastokines. The reviewers noted that the lack of mechanistic insights is a major limitation of the study (as also acknowledged by the authors).

Essential revisions:

1) The authors should show validation of the transgenic mouse in terms of the quantitative changes in Fndc5 or Irisin expression. This is important given that the in vivo data contradicts to the previously published results that reported Irisin increases bone mass and cortical bone thickness. In this study, the authors have observed a decrease in osteoblasts and an increase in osteoclasts and concomitant decrease in mineralized surface (MS/BS).

2) Another concern is related to the bone histomorphometry (Figure 4B). There is a clear reduction in BV/TV, which seems to be driven more by a reduction in the MS/BS and osteoblast numbers (p values close to 0.05) than by increased osteoclast numbers. The increase in osteoclast progenitors in the FNDC5 Tg mice is consistent with the authors' hypothesis. The authors should address this in the Discussion – perhaps the most likely explanation is that there may be a more robust increase in bone resorption and osteoclast numbers earlier in life (the mice were analyzed at 4.5 months). If data on younger mice are available, these should be provided; if not, this issue should at least be discussed.

---

## [Author Response]

Summary:This is a well-designed study of the effects of Irisin on osteoclastogenesis that reports the effects of dosing and timing of Irisin on osteoclast number, morphology, and integrin α_V_β_5_ expression in vitro. Using a combination of in vitro, RNAseq, and in vivo studies (transgenic mouse model over-expressing Fndc5, the precursor protein of irisin), the authors provide strong evidence that, in addition to its known effects on osteoblasts and osteocytes, irisin directly regulates osteoclasts. The authors also found that the hematopoietic progenitors were biased towards osteoclast differentiation along with an increase in some resorption markers and clastokines. The reviewers noted that the lack of mechanistic insights is a major limitation of the study (as also acknowledged by the authors).

The authors thank the editors and reviewers for their time in considering this manuscript. In response to the summary, we acknowledge the lack of in-depth mechanistic insights in this study but note the novelty of these findings in regard to previously uncharacterized effects of irisin on a key cell type in bone homeostasis, the osteoclast. We therefore feel that the study stands on its own while also setting the foundation for more detailed mechanistic studies that are currently in planning and underway. Below, we address each of the specific critiques in turn, with rationale and reference to corresponding changes in the revised manuscript.

Essential revisions:1) The authors should show validation of the transgenic mouse in terms of the quantitative changes in Fndc5 or Irisin expression.

We have now revised Figure 4 to more clearly demonstrate the validation of the transgenic forced expression of FNDC5 by showing quantitative increases in whole-bone mRNA expression of FNDC5 in tibia at both 4.5 and 13 months of age (Figure 4A). To establish the characterization of this transgenic model more completely we have further included data at earlier and later timepoints where available, at 2 and 13 months for µCT (Figure 4B) and at 13 months for histomorphometry (Figure 4C) (see Results and Discussion, fifth paragraph).

This is important given that the in vivo data contradicts to the previously published results that reported Irisin increases bone mass and cortical bone thickness. In this study, the authors have observed a decrease in osteoblasts and an increase in osteoclasts and concomitant decrease in mineralized surface (MS/BS).

To more fully explore the respective roles of osteoblasts and osteoclasts in this model we have included the additional histomorphometry data at 13 months of age (Figure 4C) as well as in vitro osteoblast cultures carried out from both wild type bone marrow progenitors with exogenous irisin treatment (Figure 1—figure supplement 1) and compared wild type and transgenic bone marrow progenitors (Figure 1—figure supplement 1). We have endeavored to address contradictions with previously published results throughout the Discussion (Results and Discussion, eighth paragraph).

2) Another concern is related to the bone histomorphometry (Figure 4B). There is a clear reduction in BV/TV, which seems to be driven more by a reduction in the MS/BS and osteoblast numbers (p values close to 0.05) than by increased osteoclast numbers. The increase in osteoclast progenitors in the FNDC5 Tg mice is consistent with the authors' hypothesis. The authors should address this in the Discussion – perhaps the most likely explanation is that there may be a more robust increase in bone resorption and osteoclast numbers earlier in life (the mice were analyzed at 4.5 months). If data on younger mice are available, these should be provided; if not, this issue should at least be discussed.

Thank you for that important point about the lower bone mass. We have now included µCT data for an earlier time point at 2 months, which does indeed also show a reduction in trabecular and cortical bone, and smaller bone area in transgenic mice. We have added the histomorphometry data at this earlier time point to gain insight into osteoblasts and osteoclasts relative to bone changes. In sum for the revised manuscript we included µCT at three time points (2, 4.5 and 13 months) and dynamic histomorphometry at two time points (2 months and 13 months). The original submission noted that histomorphometry was done at 4.5 months; that was in error, it was performed on 2-month-old mice and corresponds with the µCT at that time point. As noted, we have now also included in vitro osteoblast differentiation cultures of bone marrow mesenchymal progenitors from 2-3-month wild type and transgenic mice and found no increase in osteoblast differentiation or mineralization (Figure 4—figure supplement 1). Similarly, in vitro osteoblast differentiation of wild type bone marrow progenitors showed no effect of the continuous exogenous treatment with 10 ng/mL irisin that was shown to stimulate osteoclast differentiation (Figure 1—figure supplement 1).

These in vitro osteoblast findings have been included in the Discussion to inform interpretation on the respective roles on osteoblasts and osteoclasts in the transgenic model. In the transgenic model itself, the decrease in osteoblast numbers and bone formation at 2 months did not persist at 13 months, when it showed a slight but nonsignificant increase possibly related to the increase in osteoclast number (Figure 4C). At the earlier time points, osteoclast numbers did not differ by genotype, whereas by 13 months, there were more osteoclasts per bone surface in the transgenic. While the level of irisin exposure with exogenous treatment in vitro and forced FNDC5 expression in vivo are not directly comparable, these data together suggest that irisin influences both osteoblast and osteoclast differentiation throughout skeletal development. A more complete discussion of these data and the insights they may provide into the respective roles of osteoblasts and osteoclasts in response to irisin have been worked into the Discussion (Results and Discussion, eighth paragraph).